# Currents Analysis of a Brushless Motor with Inverter Faults—Part II: Diagnostic Method for Open-Circuit Fault Isolation

**Cristina Morel** [1,*] **, Baptiste Le Gueux** [2] **, Sébastien Rivero** [2] **and Saad Chahba** [1]

[1] Ecole Supérieure des Techniques Aéronautiques et de Construction Automobile, ESTACA'Lab Paris-Saclay, 12 Avenue Paul Delouvrier—RD10, 78180 Montigny-le-Bretonneux, France; saad.chahba@estaca.fr

[2] ESTACA Campus Ouest, Rue Georges Charpak—BP 76121, 53009 Laval, France; baptiste.legueux@estaca.eu (B.L.G.); sebastien.rivero@estaca.eu (S.R.)

**\*** Correspondence: cristina.morel@estaca.fr

**Abstract:** In this paper, a brushless motor with a three-phase inverter is investigated under healthy and multiple open-circuit faults. The occurrence of faults in an inverter will lead to atypical characteristics in the current measurements. This is why many usual entropies and multiscale entropies have been proposed to evaluate the complexity of the output currents by quantifying such dynamic changes. Among this multitude of entropies, only some are able to differentiate between healthy and faulty open-circuit conditions. In addition, another selection is made between these entropies in order to improve diagnostic speed. After the fault detection based on the mean values, the open-circuit faults are localized based on the fault diagnostic method. The simulation results ensure the ability of these entropies to detect and locate open-circuit faults. Moreover, they are able to achieve fault diagnostics for a single switch, double switches, three switches, and even four switches. The diagnostic time to detect and to isolate faults is between 10.85 ms and 13.67 ms. Then, in order to prove the ability of the fault diagnostic method, a load variation is performed under the rated speed conditions of the brushless motor. The validity of the method is analyzed under different speed values for a constant torque. Finally, the fault diagnostic method is independent from power levels.

**Keywords:** open-circuit switch fault; fault detection; fault isolation; fault diagnostic method



## 1. Introduction

In recent years, numerous open-switch fault diagnostic methods applied in multilevel inverters have been presented [1–3]. Fault detection and isolation methods for Mosfet or IGBT open-circuit faults have been studied in previous works, particularly focusing on arm voltages or current measurements. If open-circuit faults are not detected as quickly as possible, they may cause secondary damage. Thus, it is very important to identify and detect faulty switches as soon as possible.

Recently, many current-type methods based on the direct analysis of current waveforms have gained attention. These methods have the ability to detect and locate one or multiple open-circuit faults. Twenty-one methods for open-circuit faults were evaluated and summarized in [4], based on their performance and implementation efforts. In [5,6], a current-based method was used for fault diagnostics. The average current Park's vector strategy [5,7] was applied to diagnose the faulty upper or lower half-leg of a three-level inverter. However, the proposed strategy was not able to identify the faulty switch in the defected half-leg. In [8], a diagnostic method based on empirical mode decomposition energy entropy and normalized average current was proposed to identify one or two faults: an open-circuit fault on the upper or lower bridge arm; two open-circuit faults in the cross side bridge arm of the double phase, in the same side bridge arm of the double phase or in the double-bridge arm of the single phase. In paper [9], one open-transistor fault

was detected analyzing the normalized load current. Nevertheless, this method has a higher complexity and larger detection times. The authors of [10] proposed a fast fault detection and isolation approach to identify a single open-circuit fault in power electronics sub-modules for modular multilevel converters with model-predictive control. In [11], a comparison method of fault diagnostic variables with threshold values was presented for a three-level three-phase NPC inverter. Once a faulty leg is detected, the localization of the faulty switch is based on the value of the average current. This method can detect only single-switch faults in one time period. An implementation of open-transistor fault detection and diagnostic method based on the current trajectory of phase was presented in [12]. This method is load-independent, but detects only one open circuit. A sliding mode observer was proposed in [13–15]. The current form factors of the estimated current and measured current are needed for faulty phase detection. However, the complexity of the method is medium, and it can only detect a single switch fault. Several faults were detected in [16] using the ratio of the theoretical and the practical voltage values on the capacitor of each inverter's sub-module. The authors of [17] proposed a novel diagnostic algorithm for single and multiple IGBT open-circuit faults.

When algorithms are presented, an analysis of complexity [18] is very important. To better understand the capabilities of the algorithm, the results (under several speed values and load variations in the brushless motor) are presented in terms of computational time. The authors of [19] used a residual observer-based fault detection algorithm, detecting open-circuit faults in 15 ms to 20 ms. The authors of [20,21] proposed a Kalman-filter-based approach: the comparison of measured and estimated voltages and currents obtained by their filter enables the detection of the open-circuit faults of a modular multilevel converter sub-module. However, the diagnostic time was longer than 100 ms. In [22–24], a sliding-mode observer was proposed as a fault diagnostic method. Detecting and locating faults require at least 50 ms. In [25,26], only a single fault could be diagnosed by the state observer in more than 30 ms. A clustering algorithm and calculated capacitance methods are proposed in [27]. The algorithm is complicated and requires a large amount of calculation: both methods need at least 13 ms to detect and locate faults. An adaptive linear neuron-recursive least squares algorithm to estimate capacitor voltages is proposed to detect and locate different types of sub-module faults in [28], in more than 30 ms. The authors used simple sub-modules in [29] and integral sub-modules in [30] to detect and isolate the multiple open-switch faults in modular multi-level converters. Nevertheless, this method diagnoses and isolates multiple open-switch faults within 20 ms. The algorithm developed by [31] is based on the instant voltage error in the converter and requires only signals already available to the control system, avoiding the use of additional hardware. The algorithm is independent from the load and from the used control strategy and provides very fast detection and identification of the fault, with diagnostic times as low as two sample periods. Kiranyaz [32] used 1D CNN to detect and locate one switch open-circuit fault using circulating current, load current signals, and four-cell capacitor voltage. This method achieved a detection probability of 0.989 and an average identification probability of 0.997 in less than 100 ms. Two deep-learning methods and a stand-alone SoftMax classifier [33] were used with raw data collected by current sensors to improve classification accuracy and reduce computation time. In the method proposed by [34], the three-phase currents were used to calculate the fault diagnostic variables with the average current Park's Vector method for the identification of only one open-circuit fault. Then, these variables were processed with a fuzzy logic method [35], and the faulty information of the PWM-VSI could be obtained. Faulty power switches can be identified in less than 90 ms after fault occurrence (approximately two motor phase current fundamental periods).

Many usual entropies and multiscale entropies were proposed to evaluate the complexity of phase currents in the previous article "Current Analysis of a Brushless Motor with Inverter Faults-Part I: Parameters of Entropy Functions and Open-Circuit Faults Detection"— [36]. We are now able to select the appropriate entropy functions with an appropriate combination of parameters (as the data length, $N$; the embedding dimension,

*m*; the time lag, $\tau$; the tolerance, *r*; and the scale, *s*), which differentiate between healthy and faulty open-circuit conditions. The main goal of this paper is to present a fault diagnostic method to detect and locate the open-circuit switch faults of a brushless motor with a three-phase inverter. Among the multitude of entropies, only some are able to improve diagnostic speeds. Starting from the brushless inverter under healthy and faulty conditions, the various possible switching fault states are discussed. After the fault detection based on the mean values, the open-circuit faults are localized, based on the fault diagnostic method. The simulation results ensure the ability of these entropies to detect and locate open-circuit faults. Moreover, they are able to achieve fault diagnostics for a single switch, double switches, three switches and even four switches. Generally, the fault detection technique with more than two parameters leads to a complex detection system. That is not the case here. Fault localization is performed using a combination of entropy functions incorporating a threshold variable. Synthesis of the total computation time to detect faults in single, double, three and even four switches is realized. Then, in order to prove the ability of the fault diagnostic method, load variation (from minimum to maximum) is performed under the rated speed condition of the brushless motor. Finally, the validity of the fault diagnostic method is analyzed under different speed conditions for a constant torque.

The paper is organized as follows: Section 2 illustrates the fault location with the fault diagnostic method, together with the total computation time, to detect and locate single or multiple open-circuit switch faults. Then, the influence of torque and speed variations on the detection and location of faults is introduced in Section 3. Section 4 ends with the Conclusion.

## 2. Fault Location with the Fault Diagnostic Method

In the first part [36], many entropies were used to identify one or multiple open-circuit faults: *SampEn*, *K2En*, *CondEn*, *DispEn*, *CoSiEn*, *BubbEn*, *ApEn*, *FuzzEn*, *IncrEn*, *PhasEn*, *SlopEn*, *EnofEn* and *AttEn* functions, each one with multiscale, composite multiscale and refined multiscale approaches, providing 52 entropy functions to evaluate the complexity of the phase currents. However, among these entropies, some are more sensitive to fault conditions. Such sensitivity is able to increase the distance between the faulty and the non-faulty phases or decrease this distance in the *IncrEn* case. Some entropies will be applied to distinguish between healthy conditions and faulty open-circuit conditions, considering the following parameter settings: $N = 2000$, $m = 2$, $\tau = 1$ and $s = 2$ for $rMSBubbEn$, *PhasEn* and *CondEn*; $N = 2000$, $m = 2$, $\tau = 3$ and $s = 4$ for *SlopEn*.

Fault detection is essential in a fault diagnostic approach. Phase currents are used in a fault diagnostic procedure to detect and isolate open-circuit switches. A block diagram is presented in Figure 1. Three sensors are added to the circuit to measure the currents of the inverter ($i_a$, $i_b$ and $i_c$ of phases *a*, *b* and *c*), which can be used to identify faults in switches. In terms of algorithm complexity, the dSpace platform is suggested to implement the entropy functions due to the memory space. Furthermore, with a MATLAB Function block, we can write a MATLAB function (a functionality programmed in the M language) into a Simulink model and execute it for simulation. We specify the inputs to the MATLAB Function block in the function header as the arguments (the currents $i_a$, $i_b$ and $i_c$) and return the output data (the entropy values). In addition, after the detection of open-switch fault occurrence (Figure 2), the fault localization strategy is initiated, based on the entropy analysis of Figures 2 and 3. Moreover, the localization of the faulty switches is necessary to isolate the faults.

Fault localization is performed using one or a combination of entropy functions by incorporating threshold variables, $\epsilon_1 = 0.01$, $\epsilon_2 = 1.5$ and $\epsilon_3 = 0.2$. The selected values of $\epsilon_1$, $\epsilon_2$ and $\epsilon_3$ depend on the simulation results of the first part [36]. $\epsilon_3$ is 20 times bigger than $\epsilon_1$ and 7.5 times smaller than $\epsilon_2$.

The means of the three currents are calculated as in Figure 2. If the means of $i_a$ and $i_b$ are nearly zero, then the mean of $i_c$ is also zero. This is the normal operating condition (without faults). The phase currents of the brushless motor are alternative waves, with zero

mean. Under healthy conditions, the average value of the positive half cycle of the phase currents $i_a$, $i_b$ and $i_c$ is equal to the negative half cycle of the phase currents $i_a$, $i_b$ and $i_c$.

**Figure 1.** Schematic representation of fault diagnostic method.

The average of the phase currents $i_a$, $i_b$ and $i_c$ is nearly zero (when no fault occurs in any switch of the inverter), as in the equations

$$mean(i_a) \approx 0, \tag{1}$$

$$mean(i_b) \approx 0, \tag{2}$$

$$mean(i_c) \approx 0. \tag{3}$$

When a fault occurs in the inverter, the current waveforms are distorted. At least two of the equations are not respected: an open-switch fault in upper pair switches determines a negative current wave, with a negative mean; an open-switch fault in lower pair switches determine a positive current wave, with a positive mean. This change in the waveform carries the fault information that can be extracted using different entropy functions.

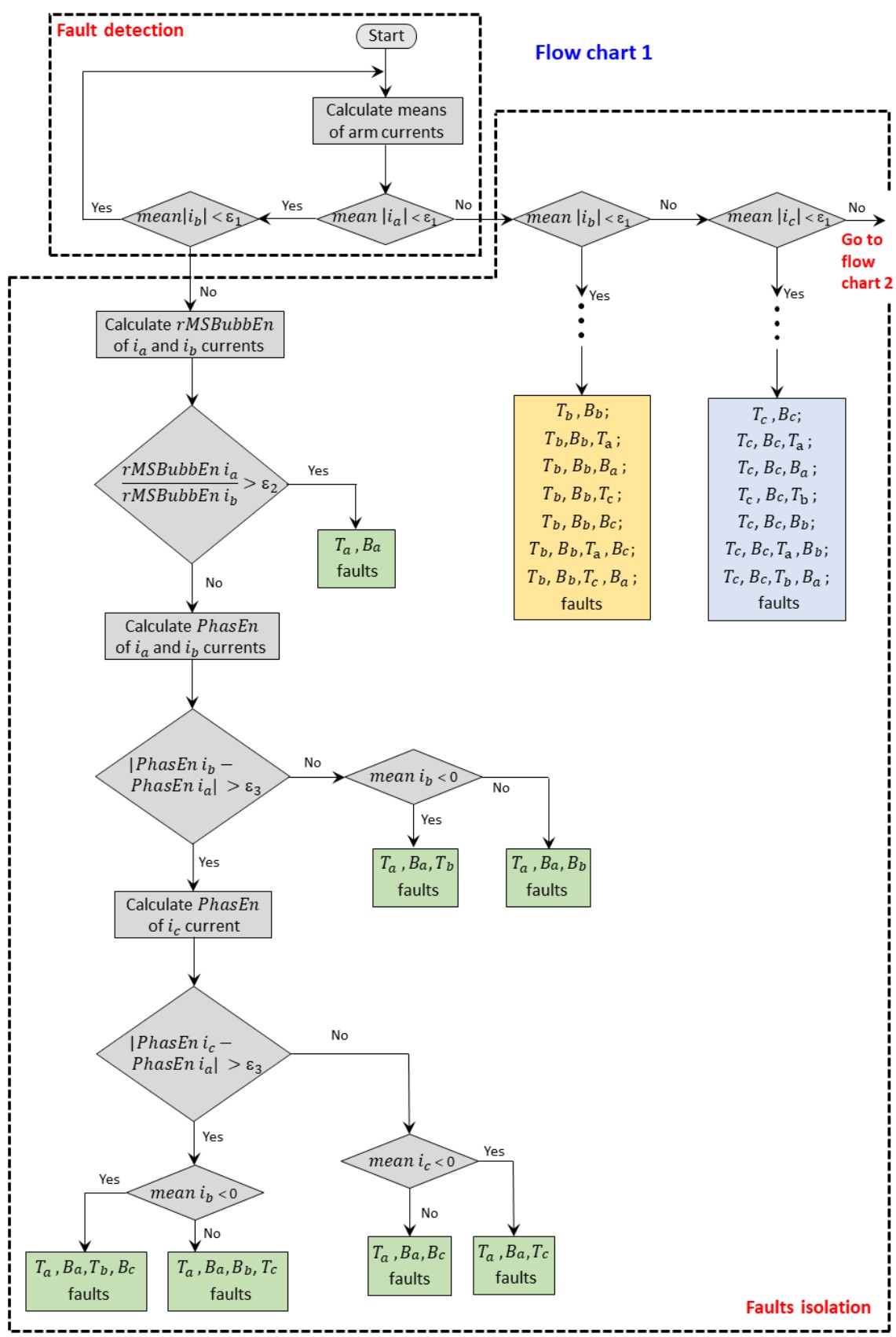

**Figure 2.** First part of the flow chart of the fault diagnostic approach.

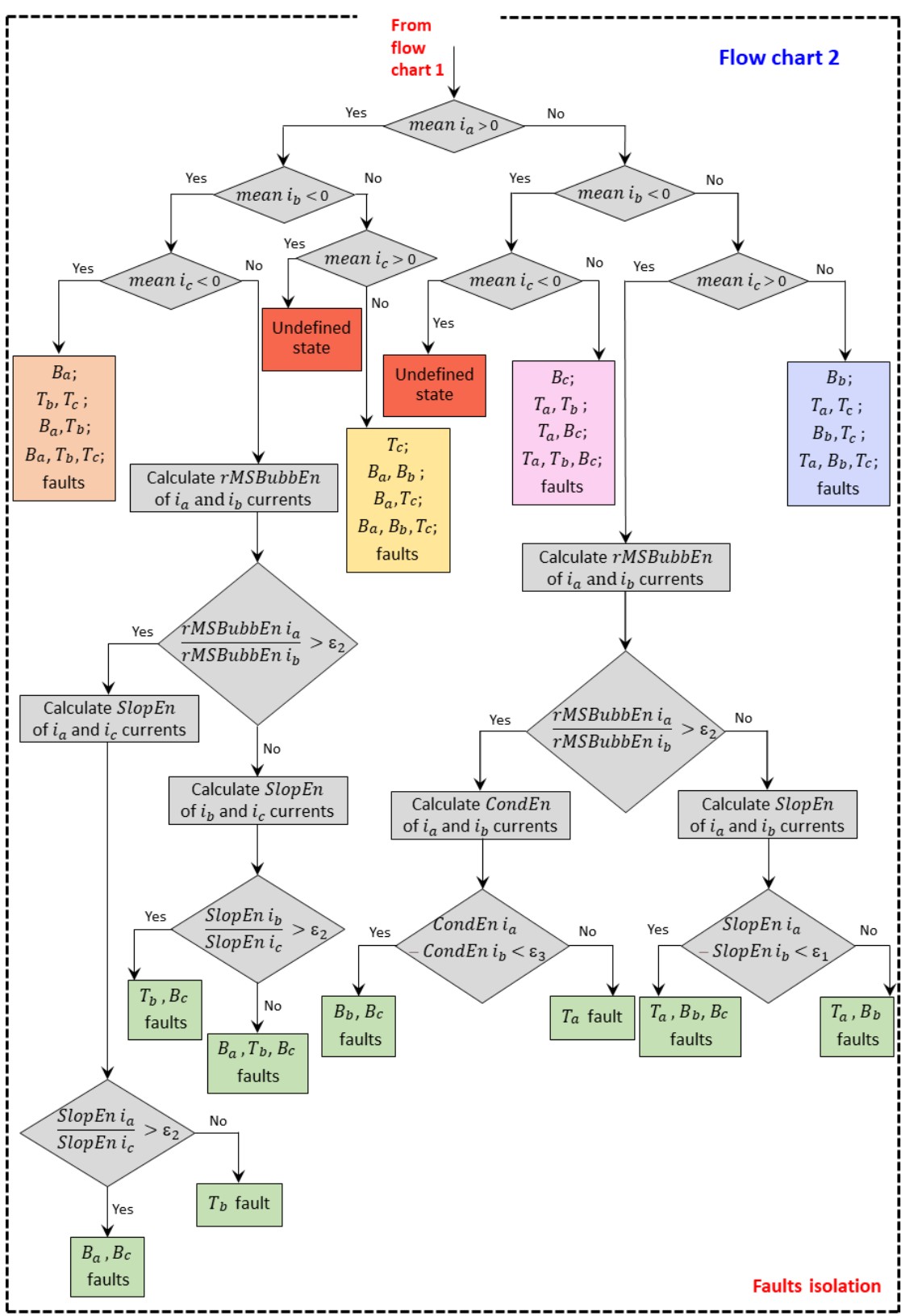

**Figure 3.** Second part of the flow chart of the proposed fault diagnostic approach.

Let us suppose that the average of the phase current $i_a$ is zero, but not for $i_b$ or $i_c$. This means that, for the moment, there are two open-circuit faults on the upper and lower switches of phase *a*. All the possibilities of open-circuit faults together with the phase current mean values are given in [36], as:

- $T_a$, $B_a$: mean of $i_a$ = 0.0006, mean of $i_b$ = 0.2708, mean of $i_c$ = −0.2714;
- $T_a$, $B_a$, $T_b$: mean of $i_a$ = 0.0004, mean of $i_b$ = −2.6371, mean of $i_c$ = 2.6367;
- $T_a$, $B_a$, $B_b$: mean of $i_a$ = 0.0044, mean of $i_b$ = 2.9920, mean of $i_c$ = −2.9964;
- $T_a$, $B_a$, $T_c$: mean of $i_a$ = 0.0025, mean of $i_b$ = 2.4690, mean of $i_c$ = −2.4715;
- $T_a$, $B_a$, $B_c$: mean of $i_a$ = 0.0032, mean of $i_b$ = −2.8272, mean of $i_c$ = 2.8304;
- $T_a$, $B_a$, $T_b$, $B_c$: mean of $i_a$ = 0.0016, mean of $i_b$ = −2.6055, mean of $i_c$ = 2.6038;
- $T_a$, $B_a$, $B_b$, $T_c$: mean of $i_a$ = 0.0041, mean of $i_b$ = 3.0170, mean of $i_c$ = −3.0211.

Once a fault is detected by the average method, the faulty switches are localized with the diagnostic method we propose, which combines the multiple entropy functions of the phase currents. The growing interest in entropy approaches relies on their ability to analyze and to provide information related to signal complexity. The entropy of the phase currents is directly used as fault information. A larger difference between the 52 entropies of phase *a* and of phases *b* and *c* is given by *SampEn*, *K2En*, *MSApEn*, *rMSBubbEn*, *FuzzEn* and *SlopEn*, as can be seen in [36]. The largest distance is obtained with *ApEn*, followed by *SampEn*. The smallest distance can be obtained with *FuzzEn* and *SlopEn*.

Another important issue for the fault detection and isolation is the assessment of computation time. Table 1 shows the computation time of all entropies. This time depends on the length of the phase current, but is not proportional to it. For a length of 6000 samples, the *SampEn* computation time is 1.44 s and 0.1862 s for a 2000-sample length. In order to reduce the fault diagnostic cost and to improve its speed, *rMSBubbEn* is chosen. This entropy is insensitive to the length of the phase current.

The *rMSBubbEn* of current $i_a$ is compared with the *rMSBubbEn* of current $i_b$. If

$$\frac{rMSBubbEn\ i_a}{rMSBubbEn\ i_b} > \epsilon_2 \tag{4}$$

is greater than the threshold $\epsilon_2$, then $T_a$ and $B_a$ switches' faults are isolated.

The computation time to calculate a mean of a 2000-sample wave is 0.2537 ms and 0.0964 ms for an *if* condition. The total computing time to detect and isolate $T_a$ and $B_a$ is 10.85 ms, for which it is necessary to calculate three means of arm currents, to apply three *if* conditions and to calculate two *rMSBubbEn* of $i_a$ and $i_b$, as in Table 2.

If Condition (4) is not respected, there is another fault on phase *b* or *c* or two faults on phases *b* and *c*. Ref. [36] shows that *PhasEn* is the right entropy to isolate the faults thereafter. *PhasEn* of the phase currents $i_a$ and $i_b$ are compared. If

$$|PhasEn\ i_b - PhasEn\ i_a| < \epsilon_3, \tag{5}$$

and the mean of current $i_b$ is negative, then there is also an open-circuit fault on $T_b$. In all, $T_a$, $B_a$, $T_b$. If the mean of the current $i_b$ is positive, then three open-circuit faults are on $T_a$, $B_a$, $B_b$. The total computing time to detect and isolate $T_a$, $B_a$, $T_b$ or $T_a$, $B_a$, $B_b$ is 12.67 ms, for which it is necessary to calculate three means of arm currents, to apply five *if* conditions and to calculate two *rMSBubbEn* and two *PhasEn* of $i_a$ and $i_b$.

**Table 1.** Entropy computation time for two different lengths of phase current.

| Entropy Type | Entropies | Computation Time for Length $L_6$ | Computation Time for Length $L_2$ |
|---|---|---|---|
| Sample Entropy | *SampEn* | 1.4441 | 0.1862 |
| | *MSSampEn* | 2.3840 | 0.2977 |
| | *cMSSampEn* | 4.1252 | 0.5084 |
| | *rMSSampEn* | 2.1296 | 0.3278 |
| Kolmokov Entropy | *K2En* | 1.5930 | 0.2009 |
| | *MSK2En* | 2.9755 | 0.3438 |
| | *cMSK2En* | 3.9457 | 0.4508 |
| | *rMSK2En* | 2.2020 | 0.2643 |
| Conditional Entropy | *CondEn* | $1.241 \times 10^{-3}$ | $7.548 \times 10^{-4}$ |
| | *MSCondEn* | $4.153 \times 10^{-3}$ | $2.916 \times 10^{-3}$ |
| | *cMSCondEn* | $18.61 \times 10^{-3}$ | $18.11 \times 10^{-3}$ |
| | *rMSCondEn* | $6.309 \times 10^{-3}$ | $4.643 \times 10^{-3}$ |
| Dispersion Entropy | *DispEn* | $2.204 \times 10^{-3}$ | $1.501 \times 10^{-3}$ |
| | *MSDispEn* | $7.120 \times 10^{-3}$ | $4.324 \times 10^{-3}$ |
| | *cMSDispEn* | $21.66 \times 10^{-3}$ | $19.28 \times 10^{-3}$ |
| | *rMSDispEn* | $9.220 \times 10^{-3}$ | $7.827 \times 10^{-3}$ |
| Cosine Similarity Entropy | *CoSiEn* | 1.801 | 0.215 |
| | *MSCoSiEn* | 2.952 | 0.327 |
| | *cMSCoSiEn* | 4.107 | 0.489 |
| | *rMSCoSiEn* | 2.467 | 0.282 |
| Bubble Entropy | *BubbEn* | $1.546 \times 10^{-3}$ | $9.429 \times 10^{-4}$ |
| | *MSBubbEn* | $10.91 \times 10^{-3}$ | $5.180 \times 10^{-3}$ |
| | *cMSBubbEn* | $18.81 \times 10^{-3}$ | $16.80 \times 10^{-3}$ |
| | *rMSBubbEn* | $7.978 \times 10^{-3}$ | $4.905 \times 10^{-3}$ |
| Approximation Entropy | *ApEn* | 2.096 | 0.260 |
| | *MSApEn* | 4.381 | 0.512 |
| | *cMSApEn* | 5.977 | 0.723 |
| | *rMSApEn* | 3.081 | 0.373 |
| Fuzzy Entropy | *FuzzEn* | 1.213 | 0.156 |
| | *MSFuzzEn* | 2.278 | 0.281 |
| | *cMSFuzzEn* | 3.196 | 0.393 |
| | *rMSFuzzEn* | 1.993 | 0.238 |
| Increment Entropy | *IncrEn* | $1.924 \times 10^{-3}$ | $1.058 \times 10^{-3}$ |
| | *MSIncrEn* | $8.639 \times 10^{-3}$ | $5.763 \times 10^{-3}$ |
| | *cMSIncrEn* | $21.42 \times 10^{-3}$ | $16.69 \times 10^{-3}$ |
| | *rMSIncrEn* | $8.931 \times 10^{-3}$ | $7.047 \times 10^{-3}$ |
| Phase Entropy | *PhasEn* | $8.168 \times 10^{-4}$ | $8.143 \times 10^{-4}$ |
| | *MSPhasEn* | $3.928 \times 10^{-3}$ | $3.159 \times 10^{-3}$ |
| | *cMSPhasEn* | $15.64 \times 10^{-3}$ | $16.62 \times 10^{-3}$ |
| | *rMSPhasEn* | $6.090 \times 10^{-3}$ | $5.229 \times 10^{-3}$ |

**Table 1.** *Cont.*

| Entropy Type | Entropies | Computation Time for Length $L_6$ | Computation Time for Length $L_2$ |
|---|---|---|---|
| Slope Entropy | *SlopEn* | $1.084 \times 10^{-3}$ | $8.216 \times 10^{-4}$ |
| | *MSSlopEn* | $5.801 \times 10^{-3}$ | $4.128 \times 10^{-3}$ |
| | *cMSSlopEn* | $18.56 \times 10^{-3}$ | $18.40 \times 10^{-3}$ |
| | *rMSSlopEn* | $7.102 \times 10^{-3}$ | $5.716 \times 10^{-3}$ |
| Entropy of Entropy | *EnofEn* | $55.20 \times 10^{-3}$ | $20.72 \times 10^{-3}$ |
| | *MSEnof*En | $139.1 \times 10^{-3}$ | $49.65 \times 10^{-3}$ |
| | *cMS*Eno*f*En | $247.6 \times 10^{-3}$ | $71.19 \times 10^{-3}$ |
| | *rMS*Eno*f*En | $118.3 \times 10^{-3}$ | $41.33 \times 10^{-3}$ |
| Attention Entropy | *AttEn* | $7.306 \times 10^{-4}$ | $7.317 \times 10^{-4}$ |
| | *MSAttEn* | $3.879 \times 10^{-3}$ | $2.768 \times 10^{-3}$ |
| | *cMSAttEn* | $14.25 \times 10^{-3}$ | $14.04 \times 10^{-3}$ |
| | *rMSAttEn* | $6.108 \times 10^{-3}$ | $5.036 \times 10^{-3}$ |

If Condition (5) is not met, and if the phase entropy of $i_a$ and the phase entropy of $i_c$ are nearby

$$|PhasEn\ i_c - PhasEn\ i_a| < \epsilon_3, \tag{6}$$

then $T_a$, $B_a$, $B_c$ faults can be isolated if the mean of current $i_c$ is positive. For negative values of $i_c$, the fault diagnostic method identifies $T_a$, $B_a$, $B_c$ as faulty. If Conditions (5) and (6) are not valid, the distances between the three *PhasEn* of $i_a$, $i_b$ and $i_c$ are greater than $\epsilon_3$, as in [36]. A positive average of $i_b$ allows the detection and isolation of $T_a$, $B_a$, $B_b$, $T_c$ faults. For a negative average of $i_b$, $T_a$, $B_a$, $T_b$, $B_c$ faults can be detected and isolated. After the calculation of three means of arm currents, applying six *if* conditions, calculation two *rMSBubbEn* and three *PhasEn*, a total computing time of 13.58 ms is obtained to detect and isolate the following cases: $T_a$, $B_a$, $T_c$ or $T_a$, $B_a$, $B_c$ or $T_a$, $B_a$, $T_b$, $B_c$ or $T_a$, $B_a$, $B_b$, $T_c$.

According to the previous analysis, when the average of $i_b$ is equal to zero and the means of $i_b$ and $i_c$ are close to zero represents a case similar to the previous one. We can isolate the following faults: $T_b$, $B_b$ or $T_b$, $B_b$, $T_a$ or $T_b$, $B_b$, $B_a$ or $T_b$, $B_b$, $T_c$ or $T_b$, $B_b$, $B_c$ or $T_b$, $B_b$, $T_a$, $B_c$ or $T_b$, $B_b$, $B_a$, $T_c$, with a total computing time as in the previous case (respecting Condition (1)).

If the mean of $i_c$ is zero and Conditions (1) and (2) are not respected, the faults $T_c$, $B_c$ or $T_c$, $B_c$, $T_a$ or $T_c$, $B_c$, $B_a$ or $T_c$, $B_c$, $T_b$ or $T_c$, $B_c$, $B_b$ or $T_c$, $B_c$, $T_a$, $B_b$ or $T_c$, $B_c$, $B_a$, $T_b$ are detected and isolated. In this case, an additional *if* condition is presented in the first flow chart of Figure 2, increasing the total computing time by 0.0964 ms.

To clearly understand the fault diagnostic method, when the averages of the currents $i_a$, $i_b$ and $i_c$ are not zero, the first flow chart of Figure 2 is extended by the second flow chart of Figure 3. Eight cases stand out in the function of $i_a$, $i_b$, $i_c$ phase current signs. If the means of these three currents are positive, negative and negative, the four cases of faults are detected and isolated as $B_a$ or $T_b$, $T_c$ or $B_a$, $T_b$ or $B_a$, $T_b$, $T_c$.

The second flow chart details the case when the averages are positive, negative and positive, highlighting $T_b$ or $T_b$, $B_c$ or $B_a$, $B_c$ or $B_a$, $T_b$, $B_c$ faults. It is impossible to have the three mean values positive, positive and positive or negative, negative and negative. If the means of these three currents are positive, positive and negative, the four faults are $T_c$ or $B_a$, $B_b$ or $B_a$, $T_c$ or $B_a$, $B_b$, $T_c$. Another possibility is negative, negative and positive, leading to the isolation of $B_c$ or $T_a$, $T_b$ or $T_a$, $B_c$ or $T_a$, $T_b$, $B_c$ faults. Details of the proposed diagnostic approach are presented in the second flow chart isolating the following faults: $T_a$ or $T_a$, $B_b$ or $B_a$, $B_c$ or $T_a$, $B_b$, $B_c$. The last case is the isolation of $B_b$ or $T_a$, $T_c$ or $B_b$, $T_c$ or $T_a$, $B_b$, $T_c$.

Let us detail the case when the averages are positive, negative and positive. The *rMSBubbEn* of current $i_a$ and $i_b$ are calculated and compared. If Condition (4) is respected,

the *SlopEn* of $i_a$ and $i_c$ is studied. In order to detect $T_b$ or $T_b$, $B_c$ or $B_a$, $B_c$ or $B_a$, $T_b$, $B_c$ faults, the following condition is applied:

$$\frac{SlopEn\ i_a}{SlopEn\ i_c} > \epsilon_2. \tag{7}$$

If this condition is true, $T_b$ is detected. Otherwise, $B_a$ and $B_c$ are isolated. Furthermore, after the calculation of three means of arm currents, applying eight *if* conditions and calculating two *rMSBubbEn* and two *SlopEn*, the total computing time is 12.84 ms for the detection and isolation of $T_b$ or $B_a$, $B_c$. If the *rMSBubbEn* of $i_a$ and $i_b$ do not fulfill Condition (4), the *SlopEn* of $i_b$ and $i_c$ are determined and compared as

$$\frac{SlopEn\ i_b}{SlopEn\ i_c} > \epsilon_2 \tag{8}$$

Providing $T_b$, $B_c$ or $B_a$, $T_b$, $B_c$ faults' isolation. The total computing time is 12.84 ms, as in the previous case.

**Table 2.** Computation time to locate faults: current phase of $L_2$ length.

| No. | Open-Circuit Fault | Number of Operation Type | Total Computation Time |
|---|---|---|---|
| 1. | No fault | 3 mean, 2 if | $8.25 \times 10^{-4}$ |
| 2. | $T_a$ | 3 mean, 8 if, 2 *CondEn*, 2 *rMSBubbEn* | $12.84 \times 10^{-3}$ |
| 3. | $T_b$ | 3 mean, 8 if, 2 *SlopEn*, 2 *rMSBubbEn* | $12.97 \times 10^{-3}$ |
| 4. | $T_c$ | 3 mean, 8 if, 2 *SlopEn*, 2 *rMSBubbEn* | $12.97 \times 10^{-3}$ |
| 5. | $B_a$ | 3 mean, 8 if, 2 *CondEn*, 2 *rMSBubbEn* | $12.84 \times 10^{-3}$ |
| 6. | $B_b$ | 3 mean, 8 if, 2 *SlopEn*, 2 *rMSBubbEn* | $12.97 \times 10^{-3}$ |
| 7. | $B_c$ | 3 mean, 8 if, 2 *CondEn*, 2 *rMSBubbEn* | $12.84 \times 10^{-3}$ |
| 8. | $T_a$, $B_a$ | 3 mean, 3 if, 2 *rMSBubbEn* | $10.85 \times 10^{-3}$ |
| 9. | $T_b$, $B_b$ | 3 mean, 3 if, 2 *rMSBubbEn* | $10.85 \times 10^{-3}$ |
| 10. | $T_c$, $B_c$ | 3 mean, 4 if, 2 *rMSBubbEn* | $10.94 \times 10^{-3}$ |
| 11. | $T_a$, $T_b$ | 3 mean, 8 if, 2 *CondEn*, 2 *rMSBubbEn* | $12.84 \times 10^{-3}$ |
| 12. | $T_b$, $T_c$ | 3 mean, 8 if, 2 *CondEn*, 2 *rMSBubbEn* | $12.84 \times 10^{-3}$ |
| 13. | $T_a$, $T_c$ | 3 mean, 8 if, 2 *SlopEn*, 2 *rMSBubbEn* | $12.97 \times 10^{-3}$ |
| 14. | $B_a$, $B_b$ | 3 mean, 8 if, 2 *SlopEn*, 2 *rMSBubbEn* | $12.97 \times 10^{-3}$ |
| 15. | $B_b$, $B_c$ | 3 mean, 8 if, 2 *CondEn*, 2 *rMSBubbEn* | $12.84 \times 10^{-3}$ |
| 16. | $B_a$, $B_c$ | 3 mean, 8 if, 2 *SlopEn*, 2 *rMSBubbEn* | $12.97 \times 10^{-3}$ |
| 17. | $T_a$, $B_b$ | 3 mean, 8 if, 2 *SlopEn*, 2 *rMSBubbEn* | $12.97 \times 10^{-3}$ |
| 18. | $T_a$, $B_c$ | 3 mean, 8 if, 2 *SlopEn*, 2 *rMSBubbEn* | $12.97 \times 10^{-3}$ |
| 19. | $T_b$, $B_a$ | 3 mean, 8 if, 2 *SlopEn*, 2 *rMSBubbEn* | $12.97 \times 10^{-3}$ |
| 20. | $T_b$, $B_c$ | 3 mean, 8 if, 2 *SlopEn*, 2 *rMSBubbEn* | $12.97 \times 10^{-3}$ |
| 21. | $T_c$, $B_a$ | 3 mean, 8 if, 2 *SlopEn*, 2 *rMSBubbEn* | $12.97 \times 10^{-3}$ |
| 22. | $T_c$, $B_b$ | 3 mean, 8 if, 2 *SlopEn*, 2 *rMSBubbEn* | $12.97 \times 10^{-3}$ |
| 23. | $T_a$, $B_a$, $T_b$ | 3 mean, 5 if, 2 *rMSBubbEn*, 2 *PhasEn* | $12.67 \times 10^{-3}$ |
| 24. | $T_a$, $B_a$, $B_b$ | 3 mean, 5 if, 2 *rMSBubbEn*, 2 *PhasEn* | $12.67 \times 10^{-3}$ |
| 25. | $T_a$, $B_a$, $T_c$ | 3 mean, 6 if, 2 *rMSBubbEn*, 3 *PhasEn* | $13.58 \times 10^{-3}$ |
| 26. | $T_a$, $B_a$, $B_c$ | 3 mean, 6 if, 2 *rMSBubbEn*, 3 *PhasEn* | $13.58 \times 10^{-3}$ |
| 27. | $T_b$, $B_b$, $T_a$ | 3 mean, 5 if, 2 *rMSBubbEn*, 2 *PhasEn* | $12.67 \times 10^{-3}$ |
| 28. | $T_b$, $B_b$, $B_a$ | 3 mean, 5 if, 2 *rMSBubbEn*, 2 *PhasEn* | $12.67 \times 10^{-3}$ |
| 29. | $T_b$, $B_b$, $T_c$ | 3 mean, 6 if, 2 *rMSBubbEn*, 3 *PhasEn* | $13.58 \times 10^{-3}$ |
| 30. | $T_b$, $B_b$, $B_c$ | 3 mean, 6 if, 2 *rMSBubbEn*, 3 *PhasEn* | $13.58 \times 10^{-3}$ |

**Table 2.** *Cont.*

| No. | Open-Circuit Fault | Number of Operation Type | Total Computation Time |
|---|---|---|---|
| 31. | $T_c, B_c, T_a$ | 3 mean, 6 if, 2 $rMSBubbEn$, 2 $PhasEn$ | $12.76 \times 10^{-3}$ |
| 32. | $T_c, B_c, B_a$ | 3 mean, 6 if, 2 $rMSBubbEn$, 2 $PhasEn$ | $12.76 \times 10^{-3}$ |
| 33. | $T_c, B_c, T_b$ | 3 mean, 7 if, 2 $rMSBubbEn$, 3 $PhasEn$ | $13.67 \times 10^{-3}$ |
| 34. | $T_c, B_c, B_b$ | 3 mean, 7 if, 2 $rMSBubbEn$, 3 $PhasEn$ | $13.67 \times 10^{-3}$ |
| 35. | $T_a, B_b, T_c$ | 3 mean, 8 if, 2 $SlopEn$, 2 $rMSBubbEn$ | $12.97 \times 10^{-3}$ |
| 36. | $T_a, B_b, B_c$ | 3 mean, 8 if, 2 $SlopEn$, 2 $rMSBubbEn$ | $12.97 \times 10^{-3}$ |
| 37. | $T_a, T_b, B_c$ | 3 mean, 8 if, 2 $SlopEn$, 2 $rMSBubbEn$ | $12.97 \times 10^{-3}$ |
| 38. | $B_a, B_b, T_c$ | 3 mean, 8 if, 2 $SlopEn$, 2 $rMSBubbEn$ | $12.97 \times 10^{-3}$ |
| 39. | $B_a, T_b, B_c$ | 3 mean, 8 if, 2 $SlopEn$, 2 $rMSBubbEn$ | $12.97 \times 10^{-3}$ |
| 40. | $B_a, T_b, T_c$ | 3 mean, 8 if, 2 $SlopEn$, 2 $rMSBubbEn$ | $12.97 \times 10^{-3}$ |
| 41. | $T_a, B_a, T_b, B_c$ | 3 mean, 6 if, 2 $rMSBubbEn$, 3 $PhasEn$ | $13.58 \times 10^{-3}$ |
| 42. | $T_a, B_a, B_b, T_c$ | 3 mean, 6 if, 2 $rMSBubbEn$, 3 $PhasEn$ | $13.58 \times 10^{-3}$ |
| 43. | $T_a, T_b, B_b, B_c$ | 3 mean, 6 if, 2 $rMSBubbEn$, 3 $PhasEn$ | $13.58 \times 10^{-3}$ |
| 44. | $B_a, T_b, B_b, T_c$ | 3 mean, 6 if, 2 $rMSBubbEn$, 3 $PhasEn$ | $13.58 \times 10^{-3}$ |
| 45. | $T_a, B_b, T_c, B_c$ | 3 mean, 7 if, 2 $rMSBubbEn$, 3 $PhasEn$ | $13.67 \times 10^{-3}$ |
| 46. | $B_a, T_b, T_c, B_c$ | 3 mean, 7 if, 2 $rMSBubbEn$, 3 $PhasEn$ | $13.67 \times 10^{-3}$ |

The second flow chart (Figure 3) provides information on the proposed diagnostic approach for the isolation of $T_a$ or $T_a$, $B_b$ or $B_a$, $B_c$ or $T_a$, $B_b$, $B_c$. After the calculation of the $rMSBubbEn$ of $i_a$ and $i_b$, $CondEn$ and $PhasEn$ are used depending on Condition (4). We present here another modality of fault identification using $CondEn$ ($SlopEn$ could also be used because they have similar compilation times). $T_a$ or $B_a$, $B_c$ are isolated if

$$CondEn \ i_a - CondEn \ i_b < \epsilon_3 \tag{9}$$

and $T_a$, $B_b$ or $T_a$, $B_b$, $B_c$ are identified if

$$SlopEn \ i_a - SlopEn \ i_b < \epsilon_1. \tag{10}$$

Finally, the fault diagnostic is complete. Table 3 illustrates the synthesis of computation time in relation to the number of faults. The shortest diagnostic time is 10.85 ms and the longest is 13.67 ms. For a typical three-phase inverter, 45 possible open-circuit faults can be diagnosed and localized with the proposed diagnostic approach, according to the flow charts of Figures 2 and 3.

**Table 3.** Synthesis computation time in relation to the number of faults.

| Number of Faults | Total Computation Time (ms) |
|---|---|
| 1 fault | 12.84–12.97 |
| 2 faults | 10.85–12.97 |
| 3 faults | 12.67–13.67 |
| 4 faults | 13.58–13.67 |

## 3. Entropy Evaluation under Load and Speed Variations

In this section, the entropy calculation has the same parameters as in Section 2, i.e., $N = 2000$, $m = 2$, $\tau = 1$ and $s = 2$ for $rMSBubbEn$ and $PhasEn$; $N = 2000$, $m = 2$, $\tau = 3$ and $s = 4$ for $SlopEn$. In the early design of the algorithm, simulation had a high significance to point out the effectiveness of the diagnostic approach. The simulation results are used for the analysis and the localization of open-circuit faults under all previously mentioned conditions. To check the independence of the method from power levels, the simulations are conducted under diverse load conditions (a variation in torque between 1 Nm and

5 Nm) with 3000 rpm constant speed. Conversely, several multiple faults are tested for a speed variation between 1000 rpm and 5000 rpm under a constant load of 3 Nm.

Figure 4 presents the simulations of $rMSBubbEn$ in the case of one open-circuit fault on $T_a$. For small values of load, $rMSBubbEn$ declines, followed by an increasing slope. At the end of the interval, $rMSBubbEn$ has a small decrease. The distance between the $rMSBubbEn$ of healthy phases (*b* and *c*), and the $rMSBubbEn$ of open-circuit phase is nearly constant no matter the load variation (Figure 4a) or speed variation (Figure 4b).

Let us take a range variation in speed within (1000 rpm to 5000 rpm) and another range for torque within (1 Nm to 5 Nm). To further illustrate the $rMSBubbEn$ for healthy phases *b* and *c* and for open-circuit phase *a*, torque curves (Figure 4a) are plotted with respect to the various speeds. The simulation results are shown in 3D graphs in Figure 5. Only for very small values of speed and torque, $rMSBubbEn$ always decreases satisfying, at the same time, Condition (4). Otherwise, the distance between the $rMSBubbEn$ of healthy phases *b* and *c* and the $rMSBubbEn$ of open-circuit phase *a* is constant. Therefore, the proposed algorithm is efficient for the entire proposed range of speed and torque, except for particular values (1 Nm and 1000 rpm).

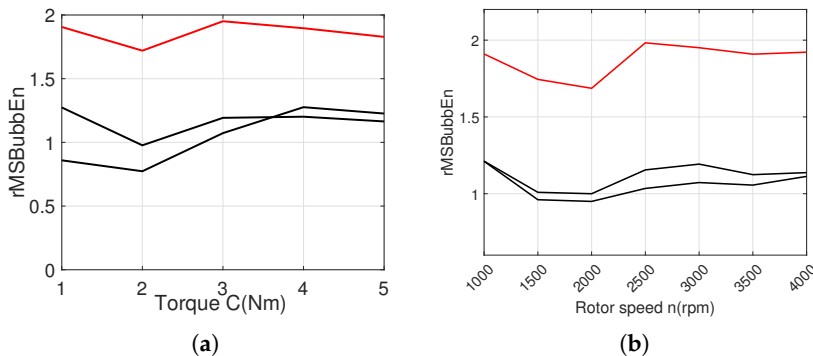

**Figure 4.** $rMSBubbEn$ for (**a**) torque and (**b**) speed variations with one open-circuit fault on $T_a$: healthy phases are represented by two black curves and the open-circuit phase is represented by a red curve.

Figure 6a displays the impact of different torque values on $rMSBubbEn$ with two open-circuits on $T_a$ and $B_a$. $rMSBubbEn$ decreases for load in the range (1, 2), increases in the range (2, 3) and is followed by a decrease in the range (3, 5). This case is similar to the previous one: the distance between the $rMSBubbEn$ of healthy phases (*b* and *c*) and the $rMSBubbEn$ of open-circuit phase *a* is nearly constant. This distance is also constant concerning the speed variations: there is a slight increase in the distance for high rotor speeds, as can be seen in Figure 6b.

Moreover, the effect of speed and torque on $rMSBubbEn$ is shown in Figure 7. The simulation results present the independence of the fault diagnostic method with different power levels for the whole ranges of speed and torque. In addition, a load of 1Nm for a speed of 1000 rpm should be avoided.

Figure 8 presents the simulation results of *PhasEn* in the case of three open-circuit faults on $B_a$, $T_b$ and $T_b$. For large values of load, the distance between the *PhasEn* of healthy phase *c* and the *PhasEn* of open-circuit phases (*a* and *b*) is nearly constant with respect to the applied speed, as in Figure 8a. Figure 8b shows that for large values of speed, this distance is invariable with respect to the applied load.

Furthermore, when the parameters *n* and *C* decrease, for small values of speed and torque, the three *PhasEn* of the phases are interweaved. A variation in *n* causes significant influence on *PhasEn* for the cases of smaller *C*, namely, for (1000 rpm and 1500 rpm) and (1 Nm and 2 Nm). Based on Conditions (5) and (6) and according to Figures 8a,b and 9, the fault diagnostic method is efficient for the ranges: (2 Nm to 5 Nm) for load variations and (1500 rpm to 4000 rpm) for speed variations.

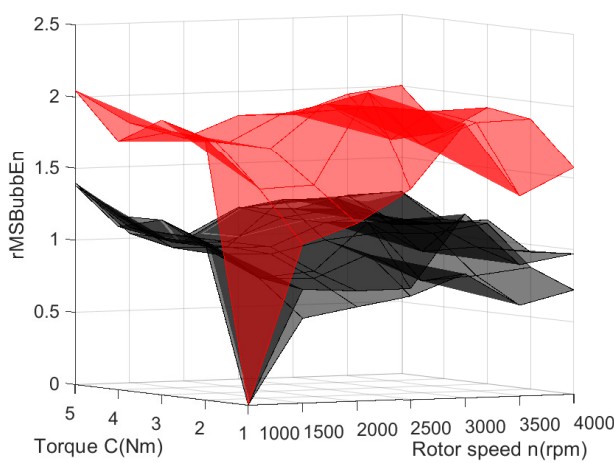

**Figure 5.** *rMSBubbEn* with one open-circuit fault on $T_a$ with different combinations of torque and speed: healthy phases are represented by two black surfaces and the open-circuit phase is represented by a red surface.

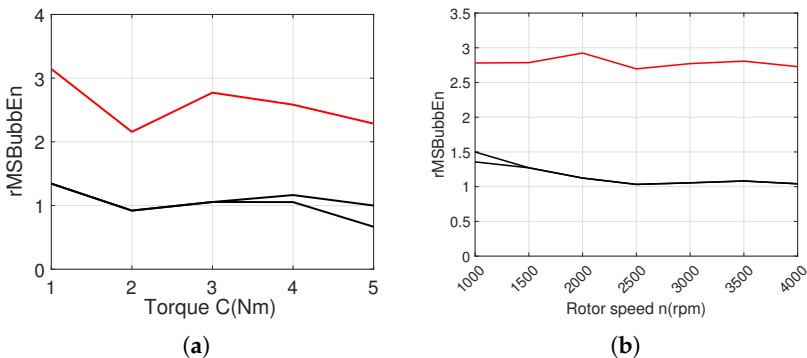

**Figure 6.** *rMSBubbEn* for (**a**) torque and (**b**) speed variations with two open-circuit faults on $T_a$ and $B_a$: healthy phases are represented by two black curves and the open-circuit phase is represented by a red curve.

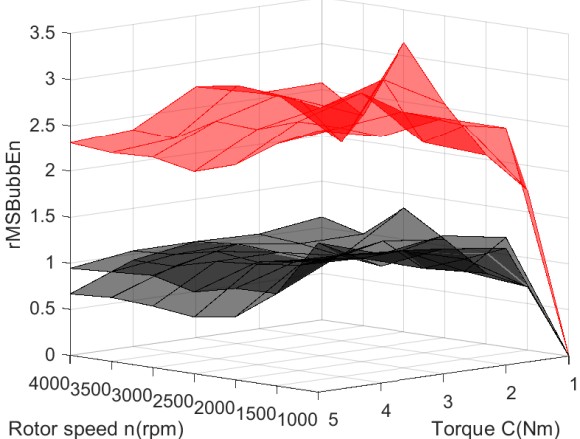

**Figure 7.** *rMSBubbEn* for torque and speed variations with two open-circuit faults on $T_a$ and $B_a$: healthy phases are represented by two black surfaces and the open-circuit phase is represented by a red surface.

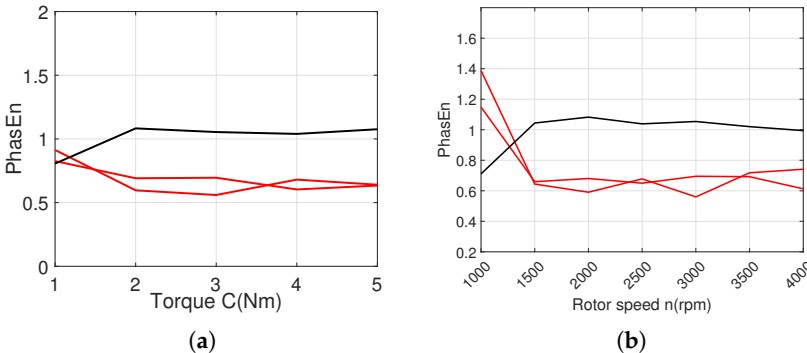

**Figure 8.** *PhasEn* for (**a**) torque and (**b**) speed variations with three open-circuit faults on $B_a$, $T_b$ and $T_b$: healthy phase is represented by a black curve and the open-circuit phases are represented by two red curves.

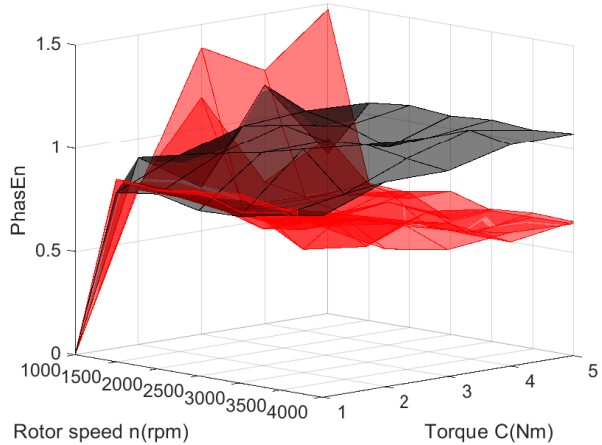

**Figure 9.** *PhasEn* for torque and speed variations with three open-circuit faults on $B_a$, $T_b$ and $T_b$: healthy phase is represented by a black surface and the open-circuit phases are represented by two red surfaces.

In order to check the incidence of the speed and load variations on *SlopEn*, Figure 10 illustrates the case of one open-circuit fault on $T_a$.

The distance between *SlopEn* of healthy phases *b* and *c* and *PhasEn* of open-circuit phase *a* is rather constant, as in Figure 10a. This distance presents a little narrowing for the load in the range of (4 Nm to 5 Nm). The simulation results underline the effectiveness of the proposed algorithm in the case of load variation in a large domain. According to Figure 10b, at the beginning of the interval (for small values of speed *n*), the three *SlopEn* of phases *a*, *b* and *c* tend to interweave.

Figure 10b shows a whole picture of *PhasEn* with various *n* and *C*. The fault diagnostic method is efficient for the range of (2000 rpm to 4000 rpm) for speed variations. As in the case above for *PhasEn* with three open-circuit faults on $B_a$, $T_b$ and $T_b$, *SlopEn* fails to isolate a $T_a$ open-circuit fault under lower speed conditions and lower torques. This is a limitation of this approach. The open-circuit faults are detected based on the mean values of phase currents, but they cannot be located only for lower values of speed and torque. On the other hand, the simulation results in Figure 11 demonstrate the effectiveness of the proposed method for high speeds (2000 rpm to 4000 rpm and high torques (2 Nm to 5 Nm).

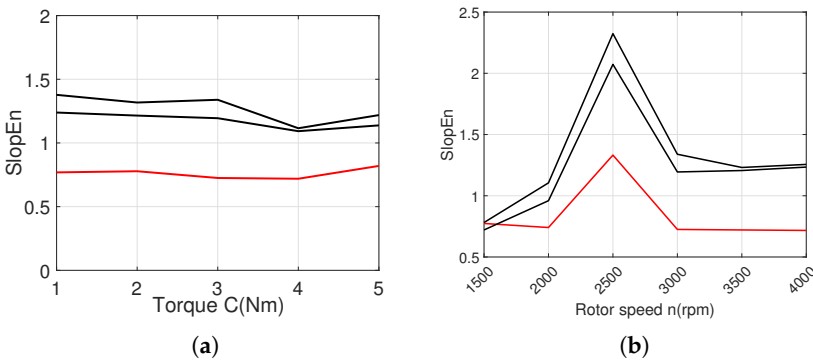

**Figure 10.** *SlopEn* for (**a**) torque and (**b**) speed variations with one open-circuit fault on $T_a$: healthy phases are represented by two black curves and the open-circuit phase represented by a red curve.

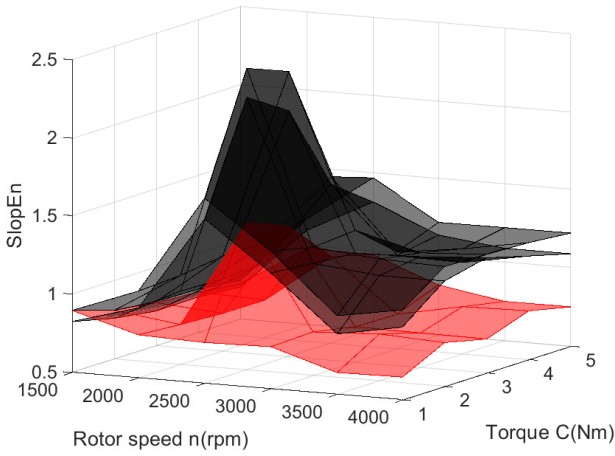

**Figure 11.** *SlopEn* for torque and speed variations with one open-circuit fault on $T_a$: healthy phases are represented by two black surfaces and the open-circuit phase is represented by a red surface.

## 4. Conclusions

Fault detection and identification are becoming increasingly important for industrial applications. This paper propose a diagnostic method for open-circuit faults of an inverter connected to a brushless motor. This algorithm requires only phase inverter currents and computing operations to generate different entropies. Some entropies are able to differentiate between healthy and unhealthy open-circuit conditions. Among these entropies, another selection is made in order to speed up the diagnostic. The simulation results ensure that these entropies are able to detect and locate open-circuit faults and, moreover, are able to achieve fault diagnostics for a single switch, double switches, three switches and even four switches.

The fault detection method is based on the average phase currents and the detection time is rather short. Then, the work in this paper deals with the localization of multiple open-circuit faults by a rapid and robust fault diagnostic method using three threshold values. The simulation results also confirm that the proposed fault diagnostic method can detect and locate multiple faults within (10.85 ms to 13.8 ms). This is much faster than many other diagnostic methods that usually require several fundamental periods. Then, in order to prove the robustness and ability of fault detection, a load variation is performed under the rated speed conditions of the brushless motor. The validity of the method is analyzed under several speed values for a constant torque.

Nevertheless, as mentioned above, there is a limit to this approach: open-circuit faults are detected, but their isolation fails under lower speed conditions and lower torques.

Consequently, the simulation demonstrates the feasibility and effectiveness of the proposed approach for large speeds of (2000 rpm to 4000 rpm) and large torques of (2 Nm to 5 Nm).

In the future, another interesting extension to our work may be to increase the brushless motor number of phases up to five (this requires an inverter with ten switches) and to compare the proposed solution with the current solution. The simulation shows that the proposed method can effectively diagnose and locate faults. Future research will focus on the proposed diagnosis method under slow variations in speed and torque (increasing and decreasing profiles), not only constant values. It will be interesting to add a fault-tolerant control strategy to our future work to ensure that the inverter and the motor work normally.

**Author Contributions:** Formulation was conducted by C.M.; problems were solved by C.M.; B.L.G. and S.R. contributed to the numerical computation and results; manuscript writing was conducted by C.M.; discussion: S.C.; revision: C.M. All authors have read and agreed to the published version of the manuscript.

**Funding:** This research received no external funding.

**Institutional Review Board Statement:** Not applicable.

**Informed Consent Statement:** Not applicable.

**Data Availability Statement:** Not applicable.

**Conflicts of Interest:** The authors declare no conflict of interest.

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
