# Peer review of "Currents Analysis of a Brushless Motor with Inverter Faults—Part II: Diagnostic Method for Open-Circuit Fault Isolation"

_actuators, doi:10.3390/act12060230_

Round 1
Reviewer 1 Report
The authors of this manuscript present and validate a fault diagnosis method to detect and locate open circuit switching faults in a brushless motor equipped with a three phase inverter. Fault detection is based on average values and open circuit fault location is possible using the fault diagnosis method. The simulation results show that the proposed method can detect and locate open circuit faults and, in addition, perform fault diagnosis for a single switch, two switches, three switches and even four switches.
The subject covered here fits perfectly into the list of topics covered by the journal Actuators. The manuscript is well structured. The written expression is mastered and the document is very pleasant to read.
However, the following shortcomings deserve to be corrected before accepting definitively the publication of this manuscript.
1. The title of the manuscript needs to be reworked. To help the authors, I suggest the following title: "Current Analysis of a Brushless Motor with Inverter Faults - Part II: Diagnostic Method for Open Circuit Fault Isolation".
2. To help the reader understand the contours of fuzzy logic, I suggest that the authors cite the following article in their literature review: https://doi.org/10.3390/en10111701
3. Figures 2 and 3 show the two parts of the flowchart for the proposed fault diagnosis approach. It would be desirable to enhance Figure 1 to indicate precisely where Figures 2 and 3 refer. Both flowcharts are substantial and it is very important for the reader to master the contours of the proposed method.
4. Please recall the conditions of the simulation at the beginning of section 3. With respect to the results presented, Figures 10 and 11 are not sufficiently analyzed. It is not at all clear what should be retained. Also in the results, there is no summary of what the reader should retain in view of the contributions listed in the general introduction.
5. In the general conclusion, the results obtained are not sufficiently quantified. In particular, I would like the authors to rewrite the second paragraph, especially when they state that the results are independent of power levels. At the end of the general conclusion, the limitations of the proposed approach should be clearly discussed. Finally, the authors should end the conclusion with a description of research perspectives.
Minor editing of English language required.
Author Response
We thank you very much for the effort put into the evaluation process and valuable comments on our manuscript, which will enable us to improve the level of scientific work. Please find below our responses to your comments.

Reviewer 2 Report
This paper has not provided any experimental or simulation waveform to validate the fault detection method. The only simulation results are just for statistic data.
The expression of this paper is also difficult to show the novelty and value of the proposed method.
The language needs further improvement.
Author Response

(The authors gave the same response as above.)

Round 2
Reviewer 2 Report
Since this paper is the second part of two series papers, the quality of this part is fine without using test or simulation waveforms. The revision of this paper is fine for me now.
The English is fine, but could be improved.